# 2INER: Instructive and In-Context Learning on Few-Shot Named Entity Recognition

**Jiasheng Zhang[1]**   **Xikai Liu[2]**   **Xinyi Lai[3]**   **Yan Gao[2]**
**Shusen Wang[2]**   **Yao Hu[2]**   **Yiqing LIN[1]**

[1]Shanghai Jiaotong University   [2]Xiaohongshu Inc.   [3]Chongqing University

{js.zhang,yiqing.lin}@sjtu.edu.cn
{xikai,yadun,haxian,xiahou}@xiaohongshu.com
laixinyi@cqu.edu.cn

## Abstract

Prompt-based learning has emerged as a powerful technique in natural language processing (NLP) due to its ability to leverage pre-training knowledge for downstream few-shot tasks. In this paper, we propose 2INER, a novel text-to-text framework for Few-Shot Named Entity Recognition (NER) tasks. Our approach employs instruction finetuning based on InstructionNER (Wang et al., 2022) to enable the model to effectively comprehend and process task-specific instructions, including both main and auxiliary tasks. We also introduce a new auxiliary task, called Type Extraction, to enhance the model's understanding of entity types in the overall semantic context of a sentence. To facilitate in-context learning, we concatenate examples to the input, enabling the model to learn from additional contextual information. Experimental results on four datasets demonstrate that our approach outperforms existing Few-Shot NER methods and remains competitive with state-of-the-art standard NER algorithms.

## 1 Introduction

Named Entity Recognition (NER) has been a fundamental task of Natural Language Processing (NLP) and there are three types of sub-tasks in NER: flat NER (Tjong Kim Sang and De Meulder, 2003), nested NER (Kim et al., 2003) and discontinuous NER (Karimi et al., 2015). All three sub-tasks aim to locate named entities, extract the entity spans, and classify each span into pre-defined label categories. In terms of the flat NER which is the main focus of this paper, it can be formulated as a sequence labeling paradigm by assigning labels to each token in the sentence through token-classification models. The dominant methods include combining Pre-trained Language Models(PLMs) (Devlin et al., 2019) with label-specific classifier (LC) (Strubell et al., 2017; Cui and Zhang, 2019). However, the fixed shape of the output LC

layer necessitates a consistent label set for both the training and testing data, which poses a challenge for knowledge transfer. Therefore, these models need to be trained from scratch to adapt to a new domain with a different label set, highlighting the requirement for a large amount of data for these methods.

Due to the high cost of sequence labeling annotation in real-world scenarios, labeled data for NER is often limited. As a result, few-shot NER has gained significant attention due to its practical applications. Meanwhile, applying prompt-base learning (Han et al., 2021) on PLMs is an effective way to solve few-shot problems (Brown et al., 2020). PLMs can learn a lot of knowledge regarding human languages by training on a large amount of self-supervised corpus. In order to explore the potential of PLMs, prompt-based learning reformulate the downstream tasks to text-to-text framework with additional prompt indicating task descriptions (e.g. instruction fine-tuning (Wei et al., 2021; Chung et al., 2022; Sanh et al., 2021)). Through this approach, the model can effectively leverage the knowledge present in PLMs to enhance downstream skills without the need for additional large amounts of downstream data. This enables the model to achieve remarkable performance in few-shot settings.

Recently, many prompt-based NER methods have emerged to address the limitations of traditional few-shot NER approaches. TemplateNER (Cui et al., 2021) treats original sentence and predicted template filled by entity spans as source and target sequence, respectively, but all candidate spans must be enumerated during inference, leading to a high computational cost. BARTNER (Yan et al., 2021) proposed a pointer mechanism to unified all NER sub-tasks into one sequence-to-sequence (seq2seq) framework. BARTNER utilizes the raw sentence as input and outputs pointer index and tag index which represent the location

of the span and the corresponding label index in the category, respectively. To further adapt BART-NER for few-shot settings, LightNER (Chen et al., 2022b) proposed a lightweight tuning approach for low-resource settings by adding a unified learnable verbalizer and incorporating learnable parameters into the self-attention layers. Nonetheless, due to the fact that pointer mechanism only outputs the indexes of entities and labels, the model encounters challenges in effectively leveraging the capabilities of PLMs to directly comprehend the semantic meaning between entities and labels. Thus instead of using a pointer mechanism, Instruction-NER (Wang et al., 2022) directly generates entity spans and types in the target sequence and applies instruction fine-tuning with two auxiliary tasks to further mining the capabilities of PLMs, which leads to significant few-shot improvements.

In terms of the auxiliary tasks in InstructionNER, they propose two auxiliary tasks from two perspectives: span recognition (Entity Extraction) and entity labeling (Entity Typing). However, we argue that NER can be further divided into three parts: 1) understand the the relationship between the label and semantic meaning of the sentence. 2) extract the spans. 3) annotate the given spans. We believe that both span recognition and entity labeling can be benefit from having a deeper understanding of the label semantics. Therefore, we proposed a new auxiliary task, called Type Extraction, to help the model to acquire this ability.

Meanwhile, none of the above methods take the additional external knowledge into account. Current literature related to utilize external knowledge in NER involve (Chen et al., 2022a) and (Lee et al., 2022a). SDNet (Chen et al., 2022a) proposes a self-describing mechanism to leverage external resources by self-describing both entity types and mentions, while (Lee et al., 2022a) uses a demonstration-based method by incorporating examples to the input but without a text-to-text framework. Therefore, to the best of our knowledge, there is currently no existing literature that combines in-context external knowledge with instruction fine-tuning for few-shot NER.

In this paper, we propose 2INER(Instructive and In-Context Learning on Few-Shot **NER**). We build upon the work of InstructionNER by incorporating in-context examples and a novel auxiliary task. Specifically, we first reformulate the NER tasks into a text-to-text framework and then employ T5

(Raffel et al., 2020) for natural language generation. In terms of the source sentence, we use instructions to distinguish between tasks by giving a comprehensive task description and include an alternative field to identify the entity type that requires detection. Moreover, we suggest incorporating in-context demonstration examples into the source sentence to enable the model to learn from external knowledge. For the target sentence, we use natural language to represent entity spans and types instead of pointer mechanism. In addition to the two auxiliary tasks used in InstructionNER, we propose a new task called type extraction to further explore the potential of PLMs to understand label semantics. Type Extraction task requires the model to identify all the entity types presented in the original sentence and learn to understand the meaning of entity types at the overall semantic level of the sentence. Our contributions can be summarized as follows:

• To utilize external knowledge, we apply demonstration-based in-context learning examples to the instruction template. The in-context examples enable the model to directly learn which spans correspond to which types from these additional information, leading to better few-shot abilities.

• We expand the NER capabilities by dividing them into three components instead of two. And we propose a novel auxiliary task for instructions fine-tuning, called type extraction, to address the existing gap. It can enable the model to understand the meaning of the entity types through the overall semantic level of the sentence, which will improve span recognition and entity labeling abilities.

• We conduct extensive experiments on four datasets, demonstrating that 2INER outperforms existing few-shot NER methods and remains competitive with SOTA standard NER algorithms.

## 2 Related Work

### 2.1 Named Entity Recognition

Currently, NER tasks can be divided into flat NER (Tjong Kim Sang and De Meulder, 2003), nested NER (Kim et al., 2003) and discontinuous NER (Karimi et al., 2015), while in this paper, we mainly focus on the flat NER task. The current dominant method to solve flat NER is using token-level classification by turning it into a sequence labeling problem (Chiu and Nichols, 2016; Liu et al., 2019; Zhang et al., 2020; Liu et al., 2021), which apply a text encoder and CRF (Ma and Hovy, 2016) in

sequence. Recently, BARTNER (Yan et al., 2021) formulate all three NER tasks into a text-to-text framework to solve them concurrently. BARTNER generate entity span sequences by a pointer-based model based on BART (Lewis et al., 2020) so that special design of tagging schema or spans post-processing are no longer needed.

## 2.2 Prompt-based Learning

With the emergence of GPT-3 (Brown et al., 2020), prompt-based learning has gained increasing attention. It can better stimulate the knowledge model learned in pre-training stages and integrate different tasks together compared to the paradigm of fine-tuning separate model for each task, especially in few-shot settings (Han et al., 2021). To push prompt-based learning further, instruction-based learning (Wei et al., 2021) is proposed to fine-tune the PLMs on a collection of task descriptions which enables the model to better follow human instructions and generalize to unseen tasks with better zero-shot and few-shot abilities (Chung et al., 2022; Sanh et al., 2021).

## 2.3 Few-Shot NER Methods

One line of work in few-shot NER is to apply contrastive learning to assign the labels by searching for the closest token (Das et al., 2022; Chen et al., 2022c), prototype (Snell et al., 2017; Fritzler et al., 2019; Ma et al., 2022b) or label semantic (Ma et al., 2022a; Huang et al., 2022) in the support set. Another line of researches is prompt-based learning using a unified text-to-text framework to make full use of the PLMs abilities. (Cui et al., 2021) applies span classification using BART and (Chen et al., 2022b; Yan et al., 2021) use a pointer mechanism to generate indexes of spans and types. (Wang et al., 2022) utilizes instruction fine-tuning and two auxiliary tasks to train T5. Meanwhile, to apply external knowledge to the model, (Chen et al., 2022a) introduces a self-describing mechanism and (Lee et al., 2022a) uses a demonstration-based method. Therefore, our methods introduce in-context learning via instruction fine-tuning together to achieve better few-shot NER abilities, which haven't been fully discussed yet in seq2seq NER settings.

## 3 Methodology

### 3.1 NER Definition

NER aims to predict all spans in the input sentence as well as the entity types associated with the spans.

The standard flatten-NER can be formulated as follows, given the input sentence containing $n$ tokens $X = [x_1, x_2, ..., x_n]$, the model have to predict the target sentence $Y = [l_1, l_2, ..., l_n]$. We use $V_{BIO}$ to denote the BIO label set, so $\forall l_i, l_i \in V_{BIO}$. While in the sequence-to-sequence modeling scenario, the input sentence is still $X$ but instead of predicting $Y$, the model predict each entity $y_i = (e_i, s_i)$ directly, where $s_i$ represents the entity span in $X$. And $e_i \in V$ represents the entity type of $s_i$, where $V$ is the set of entity types.

More specifically, we use $l$ and $r$ to indicate the left and right boundary of an entity span in $X$, so $s_i$ can be simplified as $s_i = x_{l:r}$, where $x_{l:r} = [x_l, x_{l+1}, ..., x_r]$. Therefore, the NER model have to predict each $y_i$ in $X$, indicating that the span $s_i$ belongs to the $e_i$ entity type.

### 3.2 Convert NER to Text-to-text Task

Using language models like T5 (Raffel et al., 2020) to solve most NLP tasks in a unified text-to-text framework can not only fully utilize the knowledge model learned in the pre-training stage but also simplify the training by using same data format, same loss and same model architecture. Moreover, Compared to using simple prompts, using instruction finetuning can further explore the capabilities of the model (Chung et al., 2022; Sanh et al., 2021). Besides, utilizing in-context learning can further enhance the model's few-shot capabilities in general (Brown et al., 2020) and specifically NER abilities (Lee et al., 2022b). Therefore, we transform the NER task into a text-to-text format and employ instruction finetuning and in-context learning to unleash the model's few-shot capabilities, as shown in Figure 1. The backbone we used is T5.

The basic text-to-text format of the main NER tasks consists of the following three parts, which is inspired by InstructionNER (Wang et al., 2022) [1]:

**Instruction** The instruction is a prompt that informs the model about the current task it needs to perform. The model is expected to follow the instructions provided within the prompt and complete the task accordingly. The instruction for the main NER task is: *Please extract entities and their types from the Sentence, choose entity types from Alternatives.*

**Sentence** The sentence is the input $X$ from which entities need to be extracted.

---

[1]The templates of auxiliary tasks and in-context Example will be discussed in 3.4 and 3.3 respectively.

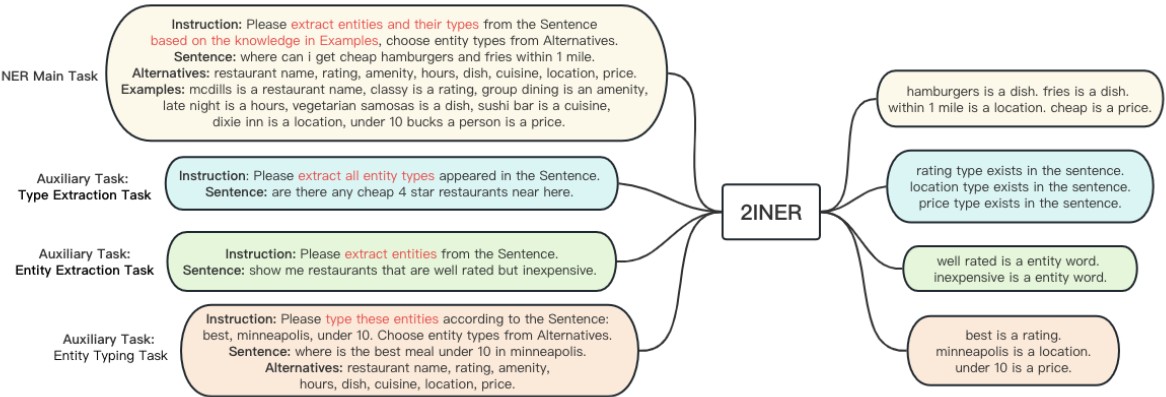

Figure 1: The model architecture of our proposed 2INER. The left and right sides are the source and target sentence of the model, respectively.

**Alternatives** Alternatives is a list of entity types ($V$) split by comma, from which the model needs to select the corresponding type to annotate the corresponding span. Alternatives serves as a constraint and a guiding factor, informing the model that it can only select entity types from within this list.

In order to formulate the NER output to natural language, for each NER output $y_i = (e_i, s_i)$, we use the following template to convert it to text: $s_i$ *is a/an* $e_i$, and we use dot to concatenate all detected entity occurrences $y_i$ to form the output text. In terms of the entity types $e_i$, we use natural language to represent the entity instead of adding special tokens to the model [2].

### 3.3 Auxiliary Tasks

To enhance the NER performance, in addition to the main task, we need to introduce several auxiliary tasks. In InstructionNER (Wang et al., 2022), they employed two auxiliary tasks: entity extraction and entity typing. Moreover, in this paper, a new auxiliary task called type extraction will be introduced. During training, the auxiliary task will also be in the form of text-to-text data, trained alongside the main task data.

The auxiliary task primarily aims to improve NER capabilities from three perspectives: understand label semantic, span recognition and entity labeling, since NER can be decomposed into three steps: understand the relationship between the label and semantic meaning of the sentence, then extract

the spans and finally annotate the given spans. We will discuss the configuration of the auxiliary task in detail from these three perspectives.

#### 3.3.1 Understand label semantic

**Type Extraction** The goal of the Type Extraction task is to identify all the entity types present in the original sentence. The Instruction is changed to: *Please extract all entity types appeared in the Sentence*. We will remove the Alternatives in this case, which means that there will be no constraints or hints regarding entity types in the input text, aiming to increase the difficulty of the task. And the output template is: $e_i$ *type exists in the sentence*. The Type Extraction task involves detecting whether a specific entity type appears in the sentence, without focusing on specific spans or associating spans with entity types. This task will assist the model in understanding the meaning of entity types at the overall semantic level of the sentence. We believe that once the model gains a deeper understanding of entity types, it will be able to comprehend the relationship between spans and types more accurately. As a result, it will enhance both span recognition and entity labeling capabilities simultaneously.

#### 3.3.2 Span recognition

**Entity Extraction** The goal of the entity extraction task is to extract useful entity spans from the original sentence without the need for annotating the extracted spans. The instruction has been modified to: *Please extract entities from the Sentence*. Because the model doesn't need to type spans, the Alternatives field is deleted. And the output template has been changed to: $s_i$ *is an entity word*, since $e_i$ is no longer needed. Because the entity ex-

---

[2]e.g. "Character _Name" will be represented as "Character Name" instead of adding a special token named "Character _Name"

traction task only require the model to predict useful spans regardless of the associated entity types, this task will guide the model to extract correct spans, enhancing the span-F1 accuracy, moreover, overall main task F1 as well (Wang et al., 2022).

The original InstructionNER (Wang et al., 2022) paper only employed span concatenation as the output(e.g. $s_1, s_2, s_3$.). However, we believe that since the output of the main task consists of complete sentences with subject-verb-object structures, it would be more cohesive to follow the same pattern for the auxiliary tasks. And more structured output can fully utilize the PLMs's understanding of the task as well.

### 3.3.3 Entity Labeling

**Entity Typing**   The entity typing task aims to type the given span with the correct label. The instruction has been modified to: *Please type these entities according to the Sentence: <the given spans>*. The Alternatives prompt and output template is the same as those in main task. During training, the given spans in the Instruction is the exact entity spans that have labels on. In entity typing task, since the spans are given, the model doesn't need to worry about the correctness of the span extracted, so the model can focus more on learning how to label the entity accurately, enhancing the main task NER ability.

### 3.4 In-Context Learning

In-context learning will be applied to further enhance few-shot NER capabilities. The main approach of in-context learning is to append Examples at the end of the input sentence, hoping that the model can directly learn which spans correspond to which types from these Examples, without the need for additional gradient updates. Besides, the in-context examples are also presented in natural language format, which closely resembles the output text format, serving as a reminder for the model about the desired format it should generate and making it easier for PLMs to understand. This similarity helps bridge the gap and facilitates the model's comprehension.

The in-context example format in NER is inspired by (Lee et al., 2022b). All examples in this context follow the template: *span is a/an entity-type*. And we will concatenate an additional prompt (*based on the knowledge in Examples*) after the Instruction to hint the model to learn from the Examples. During training stage, in-context Examples

will only be added to the main NER tasks and there will be no Examples added in auxiliary tasks, which will be discussed in detail in Analysis 5.2.

In terms of the choices of the samples in Examples, we randomly choose some spans appeared in the train set as well as their corresponding entity types to create Examples. Since we are uncertain about the entity types present in the sentence, we will provide at least one example for each entity type in the Alternatives list within the Examples. The number of samples of each entity types in Examples will also be the same [3](e.g. in terms of MIT Movie dataset, there are 12 entity types. If we set the number of examples to 5, there will be 5 examples for each entity types, resulting in a total of 5*12 examples in the field).

### 3.5 Inference

During inference time, we first use the template of the main NER task to wrap the input sentence $X$, and then feed the sentence to 2INER to get the predicted output text. In terms of the Example field, the example spans are sampled from the training support set, so the model won't see the ground-truth in the Examples during evaluation, avoiding information leakage. After the output text is generated, a decoding strategy will be applied to get the predicted entity $(e_i, s_i)$: (1) We use dot to split the whole output text to obtain individual sub-texts. (2) We use "is a" or "is an" to split each sub-text if they can be found. (3) The span is the part before "is a/an" and the entity type is the part after it. Once we get the $(e_i, s_i)$, we will check whether $s_i$ is in the input sentence $X$ and $e_i$ is in the set of entity types $V$. If it doesn't pass the check, then it isn't a valid entity and will be deleted. And if any of the three steps result in a match failure, then the sub-text will be skipped.

## 4 Experiment

### 4.1 Dataset

We conduct NER experiments in standard and low-resources settings. For the rich-resources domain, we use CoNLL-2003 (Tjong Kim Sang and De Meulder, 2003) and for the low-resource domain, we use three datasets: MIT Movie Review, MIT Restaurant Review (Liu et al., 2013) and Airline Travel Information Systems (ATIS) (Hakkani-

---

[3]We refer to "the number of samples per entity types" as "the number of examples" in the rest of the paper for convenience.

Tür et al., 2016), following (Wang et al., 2022; Chen et al., 2022b; Cui et al., 2021; Yan et al., 2021).

## 4.2 Implementation settings

In Few-Shot NER scenario, in order to guarantee that each entity type has equal number of instances in the training set, we can't sample $k$ sentences for each entity type directly because a single sentence may contain multiple entities, so the actual shot will exceed $k$. Following (Wang et al., 2022), we will apply a greedy sampling strategy (Yang and Katiyar, 2020) instead, to sample the few-shot training set for each setting and due to the randomness of the sampling, we will repeat 3 times for each experiment. We use T5-large [4] as the backbone model for fair comparision with (Wang et al., 2022). In terms of the number of examples in in-context Example field, we set the number to 5 for MIT Movie and MIT Restaurant dataset, and 1 for ATIS dataset as default [5]. We only add in-context Example field on main-task, and don't include them in auxiliary tasks. The ratio of auxiliary tasks is set to 1.0 [6]. We set the batch size to 2/4/8, learning-rate to 2e-5/5e-5 for 10/20/50 Shot settings respectively, and set batch size to 32, learning-rate to 1e-4 for the abundant data setting. The optimizer is Adam and beam search is set to 2. For evaluation, we use F1 score as the metric for NER.

The names InstructionNER in the tables mean training with main-task data only, indicating the base model, and the subscript words in the tables indicate addition to the base model: +ET, +EE, +TE, +EX means adding Entity Typing, Entity Extraction, Type Extraction, in-context examples, respectively. And we named InstructionNER$_{+ET,EE,TE,EX}$ as 2INER, which is our final model.

## 4.3 Standard NER Setting

We use CoNLL-2003 dataset to conduct standard NER experiment. We combine the train and validation set as described in (Yan et al., 2021) to train the model. The result is in Table 1, which shows that even though our method mainly focuses on few-shot NER settings, it remains competitive with

| Model | F1 | Span-F1 |
|---|---|---|
| (Yang et al., 2018) | 90.77 | - |
| (Ma and Hovy, 2016) | 91.21 | - |
| (Gui et al., 2020) | 92.02 | - |
| (Yamada et al., 2020)* | **94.30** | 92.40 |
| (Li et al., 2020a)† | - | 92.87 |
| (Yu et al., 2020a)‡ | - | 92.50 |
| LC-BERT | 91.73 | - |
| LC-BART | 90.60 | - |
| TemplateNER | 91.90 | - |
| BARTNER | - | 93.24 |
| LightNER | 92.93 | - |
| 2INER (InstructionNER$_{+ET,EE,TE,EX}$) | 90.71 | **93.93** |

Table 1: F1 and Span-F1 (%) on CoNLL-2003 Standard NER setting. Our method is competitive with SOTA algorithm and even outperform BARTNER (Yan et al., 2021) in span-F1. "*" indicates training on external data. "†" indicates the reproduction by (Yan et al., 2021). "‡" indicates the reproduction with only the sentence-level context by (Yan et al., 2021).

SOTA algorithm under standard NER setting and even outperform BARTNER (Yan et al., 2021) in span-F1, which is designed for rich-resource NER settings. The performances of 2INER in data abundant nested and discontinuous NER settings are in Appendix A.

## 4.4 Few-Shot NER Setting

Under Few-Shot NER setting, we only use K-Shot training samples to finetune our model and the results are in Table 2. According to the table, we can find that: (1) Our models consistently outperform InstructionNER as well as other baselines on all three datasets under 10/20/50 Shot settings (except 50Shot in ATIS, which is slightly lower than BARTNER). Especially in MIT Movie dataset, our models have 7.33%, 6.76%, 5.39% improvements compared to InstructionNER under 10/20/50 settings. (2) Our 10Shot model even outperforms TemplateNER's 50Shot model by 20.73% and 7.06% in MIT Movie and MIT Restaurant respectively, which highlights the superiority and capability of our model. (3) We have the same finding as InstructionNER (Wang et al., 2022) that F1 improvements are much more significant on MIT Movie than on MIT Restaurant (7.33% / 6.76% / 5.39% v.s. 6.86% / 3.24% / 3.3% under 10/20/50 Shot settings), which indicates that although MIT Movie has more entity types, text-to-text framework and instruction-tuning can better utilize pre-training knowledge, and through in-context learning, the model can learn more about the relationships between entities. (4) In ATIS dataset, the improve-

---

[4] https://huggingface.co/t5-large

[5] ATIS has 79 entity types so we set the number to 1 to avoid excessively long token lengths.

[6] The data size ratio between main task and each auxiliary tasks. 1.0 means that each sample will be extended into 4 samples: one for main task, one for EE, ET, TE, respectively.

| Models | MIT Movie | | | MIT Restaurant | | | ATIS | | |
|---|---|---|---|---|---|---|---|---|---|
| | 10 | 20 | 50 | 10 | 20 | 50 | 10 | 20 | 50 |
| LC-BERT | 25.2 | 42.2 | 49.6 | 21.8 | 39.4 | 52.7 | 44.1 | 76.7 | 90.7 |
| LC-BART | 10.2 | 27.5 | 44.2 | 6.3 | 8.5 | 51.3 | 42.0 | 72.7 | 87.5 |
| TemplateNER | 37.3 | 48.5 | 52.2 | 46.0 | 57.1 | 58.7 | 71.7 | 79.4 | 92.6 |
| BARTNER* | 41.1 | 54.0 | 67.7 | 44.0 | 56.0 | 64.0 | 77.7 | 86.1 | **93.4** |
| LightNER | 41.7 | 57.8 | 73.1 | 48.5 | 58.0 | 62.0 | 76.3 | 85.3 | 92.8 |
| InstructionNER | 64.4 (±2.1) | 70.0 (±0.3) | 74.1 (±1.2) | 58.7 (±1.2) | 65.5 (±1.4) | 71.2 (±1.1) | 90.14 (±0.12)† | 91.22 (±0.19)† | 92.53 (±0.14)† |
| InstructionNER$_{+ET,EE}$ | 65.6 (±3.0) | 70.1 (±1.9) | 74.7 (±0.3) | 58.9 (±0.8) | 66.1 (±0.9) | 71.1 (±0.9) | 90.04 (±0.02)† | 91.46 (±0.23)† | 92.62 (±0.04)† |
| InstructionNER$_{+EX}$ | 72.56 (±1.01) | 74.99 (±0.27) | 78.61 (±0.37) | 64.07 (±1.25) | 68.2 (±0.11) | 74.38 (±0.19) | 89.17 (±0.2) | 91.33 (±0.05) | 92.65 (±0.18) |
| InstructionNER$_{+TE}$ | 72.0 (±0.25) | 76.55 (±0.2) | 80.02 (±0.26) | 65.52 (±1.35) | 68.67 (±0.95) | 73.98 (±0.27) | **90.77 (±0.6)** | 91.85 (±0.05) | 92.69 (±0.1) |
| InstructionNER$_{+ET,EE,TE,EX}$ | **72.93 (±0.91)** | **76.86 (±0.53)** | **80.09 (±0.22)** | **65.76 (±0.47)** | **69.34 (±0.81)** | 74.4 (±0.4) | 90.47 (±0.26) | **92.11 (±0.09)** | 92.83 (±0.15) |

Table 2: The F1(%) on three dataset under 10/20/50 Shot settings. The **bold** number means the best F1 across all models and the numbers in brackets means the standard deviation. The underline numbers mean the best results in our experiments. The "†" numbers mean the results of our reproduction. "*" means the reproduction by InstructionNER (Wang et al., 2022).

ment of our model is less significant compared to other two dataset. We argue that this is because ATIS contains 79 entity types and even if we only provide one sample span for each entity types in in-context Example field, the average token length is 1099 compared to 368 with or without examples, where the token length of the Alternative filed is 327. So the actual input Sentence $X$ only accounts for 3.7% of the total token length, which increases the difficulty for the model to extract key information from lengthy sentences. [7] So too many entity types may potentially reduce model improvements.

## 4.5 Ablation Study

In order to find out the influence of our proposed type extraction task and in-context examples on model's few-shot abilities, we conduct ablation studies in Figure 2. The results indicate that adding type extraction task and in-context examples can further enhance the model's few-shot NER abilities. We set InstructionNER as the baseline here which only trains on main-task data without any auxiliary tasks. Then we add type extraction task (InstructionNER$_{+TE}$) or in-context examples (InstructionNER$_{+EX}$) respectively on the baseline model to explore their influences. The results from Figure 2 shows that in terms of 10/20/50 Shot settings in few-shot NER, type extraction task achieves an average improvements of 7.21%, 4.86%, 4.35% F1 and in-context example achieves an average improvements of 6.76%, 3.84%, 3.84% F1 in MIT Movie and MIT Restaurant dataset.

Moreover, adding type extraction task can greatly increase the Span-F1 as well. Because Span-F1 indicates the model's ability to locate

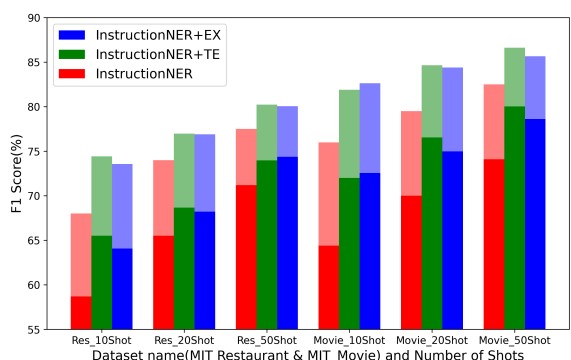

Figure 2: F1 and Span-F1 (%) on MIT Movie and MIT Restaurant through 10/20/50 Shot settings with different task combinations. The deep and light color indicate F1 and Span-F1 respectively.

spans, the results reveal that through training on type extraction task, span recognition can be benefit from having a deeper understanding of the labels from the overall semantic level of sentence. Therefore, it proves the effectiveness of three steps of NER abilities we proposed in 3.3, and shows that type extraction task can simultaneously improve span recognition and entity labeling abilities through understanding label semantic.

## 5 Analysis

### 5.1 Increase Example Number

In this section, we will focus on how the number of examples in in-context Example field influence the model performance. We will sequentially change the number of examples to 1, 3, 5, 10, and 15, and train corresponding models to observe the change of F1 on MIT Restaurant dataset. We train our model with main-task and in-context example without any auxiliary tasks (InstructionNER$_{+EX}$) in this section. The results are in Table 3.

---

[7] We try to use special-tokens to represent the entity types, but the F1 is slightly lower than without using special-tokens and the proportion of $X$ to the total number of tokens is 4.5%.

| InstructionNER$_{+EX}$ | MIT Restaurant | |
| --- | --- | --- |
| Examples | 20 Shot | 50 Shot |
| 0 | 65.5 (±1.4) | 71.2 (±1.1) |
| 1 | 67.74 (±0.22) | 73.89 (±0.15) |
| 3 | 67.89 (±0.3) | 74.15 (±0.39) |
| 5 | 68.2 (±0.11) | 74.38 (±0.19) |
| 10 | 69.47 (±0.35) | 74.41 (±0.18) |
| 15 | **69.52 (±0.16)** | **74.64 (±0.49)** |

Table 3: F1 scores(%) on MIT Restaurant dataset while changing number of examples using InstructionNER$_{+EX}$. **Bold** numbers indicate the best F1 and the numbers in brackets means the standard deviation.

As the number of examples increases, F1 score continues to increase and the largest improvement in F1 score occurs when going from zero examples to one example. As the number of examples increases further, the F1 will continue to increase but the rate of improvement gradually slows down. This suggests that when only one in-context example is provided, the model can quickly learn the specific meanings of each entity type from the example. While more examples may lead to repetitive cues to the model so a balance should be made between model performance and computational cost.

## 5.2 Effect of In-Context Example on Auxiliary task

In this section, we will discuss whether to add in-context examples on auxiliary task. The model is 2INER (InstructionNER$_{+ET,EE,TE,EX}$) and we will compare two settings: add examples only on main-task, add examples on main-task as well as three auxiliary tasks. The results in Table 4 indicates that adding examples on auxiliary task will slightly decrease the F1 performance. Because adding examples to auxiliary tasks may potentially reduce their difficulty and make it too easy for the model, thereby reducing the auxiliary tasks' effectiveness in aiding the main task. So adding examples only to the main task is a better approach.

## 5.3 Increase Shot

In this section, we will discuss the model performance under relatively abundant settings. We increase the shots to 100, 200 and 500 in MIT Movie and MIT Restaurant datasets using 2INER (InstructionNER$_{+ET,EE,TE,EX}$). As shown in Table 5, compared to InstructionNER, 2INER achieves 5.43%, 3.98%, 3.19% improvements in F1 under 100/200/500 shots settings respectively.

| | MIT Restaurant | | |
| --- | --- | --- | --- |
| | 10 Shot | 20 Shot | 50 Shot |
| 2INER | 65.26 | 69.27 | 74.2 |
| Examples on all tasks | (±0.49) | (±0.89) | (±0.45) |
| 2INER | **65.76** | **69.34** | **74.4** |
| Examples only on Main-Task | (±0.47) | (±0.81) | (±0.4) |

Table 4: The comparison between adding in-context examples only on main-task and on all tasks including auxiliary tasks. **Bold** numbers indicate the best F1 and the numbers in brackets means the standard deviation.

| Models | MIT Movie | | | MIT Restaurant | | |
| --- | --- | --- | --- | --- | --- | --- |
| | 100 | 200 | 500 | 100 | 200 | 500 |
| LC-BERT | 50.7 | 59.3 | 74.4 | 53.5 | 57.4 | 61.3 |
| LC-BART | 47.5 | 54.2 | 64.1 | 52.2 | 56.3 | 60.2 |
| TemplateNER | 56.3 | 62.0 | 74.9 | 60.1 | 62.8 | 65.0 |
| BARTNER* | 70.1 | 74.6 | 82.6 | 65.3 | 74.4 | 75.7 |
| LightNER | 78.0 | 80.6 | 84.8 | 70.8 | 75.5 | **80.2** |
| InstructionNER$_{+ET,EE}$ | 74.3 | 78.4 | 82.3 | 72.7 | 75.5 | 76.6 |
| 2INER | **81.3** | **83.54** | **86.16** | **76.57** | **78.31** | 79.11 |

Table 5: The F1 (%) under relatively abundant settings. "*" indicates the reproduction results by (Wang et al., 2022). **Bold** numbers indicate the best F1.

And 2INER outperforms LightNER in all settings except 500-shots in MIT Restaurant, which shows that 2INER has great NER abilities under data abundant scenario as well. We argue that the in-context Example field may help the model to learn from more diverse samples from the abundant training set and turn the general knowledge into specialized capabilities, leading to the improvement in F1.

## 6 Conclusion

In this paper, we propose 2INER for few-shot NER using both instruction finetuning and in-context learning by converting NER into a text-to-text framework. Based on InstructionNER, we create a template to concatenate task-specific instructions, input sentence and entity alternatives to make full use of the pre-training knowledge. Besides, we decompose NER into three steps and introduce another auxiliary tasks, called type extraction, to help the model better understand the general semantic meaning of the entity types, which can improve both span recognition and entity labeling abilities. Moreover, we apply the in-context examples to enable the model to learn from additional contextual information, enhancing few-shot abilities. Multiple experiments on four NER datasets prove 2INER's effectiveness in few-shot NER scenario by consistently outperforming other baselines.

## Limitations

One limitation of our work is the extensive length of the Example and Alternative field when there are too many existed entity types. While incorporating in-context examples in the input sentence can improve few-shot NER performance, it poses a challenge when the Example field becomes too long because we add at least one examples for each potential entity type, especially when the Alternative list contains numerous entity types. This can result in less improvement gains and more computational costs. To address this issue, we assume that larger PLMs such as the recently proposed LLaMA (Touvron et al., 2023) could potentially be explored in future research as a means of resolution.

## Ethics Statement

In consideration of ethical concerns, we would make the following descriptions: (1) All of our experiments are conducted using existing datasets sourced from publicly available scientific papers. (2) Our few-shot methods don't require a lot of computational resources. (3) Our text generation models will generate texts based on existing templates, so it won't generate harmful sentences.

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

# A   Appendix

In this section, we will discuss the remaining two NER settings: nested NER and discontinuous NER. Because the text-to-text structure of our proposed method can be easily adapted to all three NER settings, which will result in a unified structure for solving NER problems. Here, we mainly discuss standard NER scenarios with abundant data.

For data abundant nested NER, We conduct experiments on Genia (Kim et al., 2003). We follow BARTNER (Yan et al., 2021) to use five entities types and split the train, dev, test as 8.1:0.9:1.0. The results are in Table 6.

For data abundant discontinuous NER, we conduct experiments on CADEC (Karimi et al., 2015).

| Genia: Model | P | R | F |
|---|---|---|---|
| (Li et al., 2020b)[BERT-Large]† | 81.25 | 76.36 | 78.72 |
| (Yu et al., 2020b)[BERT-Large]† | 79.43 | 78.32 | 78.87 |
| (Wang et al., 2020)[BERT-Large] | 79.45 | 78.94 | 79.19 |
| BARTNER (Yan et al., 2021) | 78.87 | 79.6 | 79.23 |
| 2INER | **82.9** | **80.74** | **81.81** |

Table 6: Span-F1 (%) on Genia Nested data abundant NER setting. The "†" mean the reproduction by (Yan et al., 2021).

| CADEC: Model | P | R | F |
|---|---|---|---|
| (Metke-Jimenez and Karimi, 2016) | 64.4 | 56.5 | 60.2 |
| (Tang et al., 2018) | 67.8 | 64.9 | 66.3 |
| (Dai et al., 2020)[ELMo] | 68.9 | 69.0 | 69.0 |
| BARTNER (Yan et al., 2021) | 70.08 | 71.21 | 70.64 |
| 2INER | **71.18** | **75.26** | **73.16** |

Table 7: Span-F1 (%) on CADEC discontinuous data abundant NER setting.

Following BARTNER (Yan et al., 2021), since only the Adverse Drug Events (ADEs) entities include discontinuous data, only these entities were considered. The results are in Table 7.

The experiment settings are the same as flat NER. We use T5-large as the backbone model and report span-level F1. The results show that in data abundant nested and discontinuous NER setting, our proposed method greatly outperforms BARTNER (Yan et al., 2021) and other SOTA methods, which demonstrates that our methods do have a potential to handle different NER settings in a unified framework.