# OpenReview forum: "2INER: Instructive and In-Context Learning on Few-Shot Named Entity Recognition"
_EMNLP/2023/Conference — EMNLP 2023 Findings_

### Official Review · Reviewer_aEJr · 2023-08-01

**Soundness:** 3

**Excitement:**

3: Ambivalent: It has merits (e.g., it reports state-of-the-art results, the idea is nice), but there are key weaknesses (e.g., it describes incremental work), and it can significantly benefit from another round of revision. However, I won't object to accepting it if my co-reviewers champion it.

**Paper Topic And Main Contributions:**

The paper presents a text-to-text approach for Named Entity Recognition, extending InstructionNER by (i) adding an additional auxiliary task that tasks a model to predict the existence of entity types in an input, (ii) adding in-context learning in addition to fine-tuning. Experiments on three datasets in a few-shot setting indicate performance improvements over InstructionNER and all other few-shot baselines. An ablation study indicated the beneficial properties of both the auxiliary task as well as adding additional in-context learning.

**Questions For The Authors:**

- What distinguishes 2INER+EX from 2INER? Or rather, is 2INER without +EX  equal to InstructionNER
-  Following this question, how is it then possible that 2INER+TE and 2INER+EX individually improve so much over InstructionNER, yet combining them only improves marginally over each modification individually? Have the authors looked into that?

**Reasons To Accept:**

- The modifications the authors made to InstructionNER improved results substantially on two out of three datasets
- Ablation study and Analysis to understand which modification impacts performance and to which extent.
- Results improve results beyond few-shot classification to many samples (tested up to 500), albeit the performance gap appears to decrease (Table 5, 500 samples MIT Restaurant)

**Reasons To Reject:**

- The modifications made to InstructionNER are relatively minor and relatively straightforward additions. The new auxiliary loss is essentially the counterpart of the entity extraction loss for the entity types.
- The experimental setup is confusing. Why are the authors using a different name for 2INER and InstructionNER when they are identical, apart from the subscripts added to them which specify each modification (based on the observation that Table 3, 0 2INER in-context examples matches InstructioNER results in Table 2)? Furthermore, the setup around the added in-context examples is not entirely clear to me: are the in-context examples different from the samples used for few-shot fine-tuning (this is what Table 3 suggests)? In that case, the performance comparison in Table 2 between 2INER and the other models would not be sound. Can the authors provide more details here?

**Reproducibility:**

4: Could mostly reproduce the results, but there may be some variation because of sample variance or minor variations in their interpretation of the protocol or method.

**Reviewer Confidence:**

3: Pretty sure, but there's a chance I missed something. Although I have a good feel for this area in general, I did not carefully check the paper's details, e.g., the math, experimental design, or novelty.

**Typos Grammar Style And Presentation Improvements:**

- The paper can be written much more concisely. The authors do not very clearly distinguish the paper's contributions from the methodology of InstructionNER. For instance, mentioning flat/nested NER at the very beginning of the introduction (and in the related work section later on) is confusing -- the authors do not consider different NER settings in their paper. The following two chapters continue to not be directly relevant to the paper, making the paper difficult to follow. It is not until l.101 that the content becomes directly relevant to the paper's contribution. The authors should consider revising every section, and asking for every sentence if it is relevant to the reader.
- Type Extracting -> Type Extraction
- A Figure showing text-to-text NER in the introduction would be nice

---

> ### Author Rebuttal · Authors · 2023-08-28
>
> > The modifications made to InstructionNER are relatively minor and relatively straightforward additions. The new auxiliary loss is essentially the counterpart of the entity extraction loss for the entity types.
>
> We admit that our modifications are not huge progresses based on InstructionNER. But the added Type Extraction task indeed help both Entity Typing and Entity Extraction abilities, and results in a great improvement in Few-Shot NER F1 scores.
>
> Furthermore, if we think of auxiliary tasks as a thinking process for completing the NER task，they are similar to chain of thought.
>  - Step1 - Type Extraction: The Step 1 is to understand the meaning of the entity types through the overall semantic level of the sentence.
>  - Step2 - Entity Extraction: Based on the label semantic, the Step 2 is to decide which span to extract. (If we have a deeper understanding of label semantic of entity types, we can focus more on the required spans instead of all spans, simplifying the task that the model needs to perform. And knowing the entity type better can absolutely help the model to extract related spans easier.)
>  - Step3 - Entity Typing: After extract the span, the Step 3 is to properly typing these span (If the span extracted in Step 2 is more accurate, the typing will be easier).
>
> The difference between these steps and chain of thought(COT) is that:
>  - COT directly generate the thinking process by the model itself, because different tasks require different thinking process.
>  - Because the thinking process of NER can be manually constructed, we can use auxiliary tasks to represent the chain of thought process. So that we can use gradient to update the parameters to directly ask the model to follow our constructed thinking process.
>  - So there is no need to generate the COT process in the model output, because these knowledge has been updated by the gradient.
>
> So we believe that the Type Extraction is not just the counterpart of Entity Extraction. It is the first step in the decomposed chain of thought process to solve the overall NER problem because understanding label semantic is crucial.
>
>
> > The experimental setup is confusing. Why are the authors using a different name for 2INER and InstructionNER when they are identical, apart from the subscripts added to them which specify each modification (based on the observation that Table 3, 0 2INER in-context examples matches InstructioNER results in Table 2)?
>
> > What distinguishes 2INER+EX from 2INER? Or rather, is 2INER without +EX equal to InstructionNER
>
> I'm very sorry that this error has caused you trouble. And we admit that "InstructionNER" == "2INER" is confusing.
>
> In line-470, I explained that "The names InstructionNER and 2INER in the tables mean training with main-task data only". The intention of my statement is that the meanings of "InstructionNER" and "2INER" in the table are equivalent ("InstructionNER" == "2INER").
>
> In other words, the meaning of the symbol can be simplified as follows:
>
> | Symbol                       | Training Tasks Details                                                                |
> |------------------------------|---------------------------------------------------------------------------------------|
> | InstructionNER & 2INER       | Main-Task                                                                             |
> | 2INER+EX          | Main-Task + Demonstrate Example                                                       |
> | 2INER+TE          | Main-Task + Type Extraction                                                           |
> | 2INER+ET,EE,TE,EX | Main-Task + Entity Typing + Entity Extraction + Type Extraction + Demonstrate Example |
>
> Again, we are truly sorry for the trouble in our symbol. This confusing symbols will be modified. Thank you for pointing that out!
>
> > Furthermore, the setup around the added in-context examples is not entirely clear to me: are the in-context examples different from the samples used for few-shot fine-tuning (this is what Table 3 suggests)? In that case, the performance comparison in Table 2 between 2INER and the other models would not be sound. Can the authors provide more details here?
>
> We admit that the setup about in-context examples may not be very clear and I will discuss it more precisely.
>
> The setup:
>  - Training: During instruction fine-tuning, all the in-context examples we concatenated in the input are from few-shot support set. In other words, if the K-shot is 20 and we have 4 entity types, then we will have 20 * 4 = 80 examples in total ideally. And if we use 5 in-context examples during fine-tuning, 5 random examples will be sampled from 80 support set to formulate the in-context learning materials. And for each sample in training set, we will sample different 5 examples.
>  - Inference: During inference, the in-context examples are from training support set (which is 80 examples in total). And for each sample in evaluation set, random 5 in-context examples will be sampled from those 80 support set.
>  - To sum up, all instances in in-context examples are sampled from training support set. Therefor, there will be no information leakage during evaluation.
>
> Additionally, in that case, in Table-3, we only do experiments under 20-Shot and 50-Shot settings without 10-Shot, because we don't have enough 15 in-context examples under 10-Shot setting. (only have 10 examples / entity types)
>
> We apologize for not have clearer explanations related to these setups in our paper.
>
> > Following this question, how is it then possible that 2INER+TE and 2INER+EX individually improve so much over InstructionNER, yet combining them only improves marginally over each modification individually? Have the authors looked into that?
>
> Yes, it is indeed a good question why the improvement combining them is relatively smaller. We believe that the reason may be the lack of data. As we mainly focus on few-shot settings, the highest level of performance that a model can achieve is limited by the amount of data available. So maybe the model has already hitting the upper boundary in two MIT dataset under few-shot settings given the sacle of the model(T5-large). And in ATIS dataset, the combining improvement and individual improvement are rather equivalent, so this phenomenon only exists in two MIT dataset.
>
> Therefore, we think it is reasonable to believe that small few-shot data scale and model scale limit the potential of the combining performance.
>
> > The paper can be written much more concisely. The authors do not very clearly distinguish the paper's contributions from the methodology of InstructionNER. For instance, mentioning flat/nested NER at the very beginning of the introduction (and in the related work section later on) is confusing -- the authors do not consider different NER settings in their paper. The following two chapters continue to not be directly relevant to the paper, making the paper difficult to follow. It is not until l.101 that the content becomes directly relevant to the paper's contribution. The authors should consider revising every section, and asking for every sentence if it is relevant to the reader.
>
> We admit that the structure of the introduction may be confusing. And we have taken your suggestions into account to revise the paper.
>
> The reason why we discuss different NER settings (doing main experiments on flatten NER, but discuss nested / discontinuous NER in introduction) is because we believe that our method can be easily adapted to all three NER settings, which will result in a unified structure for solving both NER and few-shot NER problems.
>
> But we didn't conduct experiments in these two settings. And here is the reason we didn't include nested / discontinuous experiment in our paper:
>  - As we mainly focus on few-shot settings, current literatures mainly discuss few-shot NER under flatten scenario, and there are little papers discuss nested / discontinuous few-shot settings. So it is difficult to find comparisons for them if we focus on few-shot.
>  - The reason why little literature discuss few-shot nested / discontinuous NER is that nested / discontinuous instances only accounts for a small proportion. For example, CADEC [1](Karimi et al.,2015) (discontinuous) only have 10.6% of the mentions are discontinue and GENIA [2](Kim et al.,2003) (nested) only have 8.6% of the mentions are nested. So there will not be enough nested / discontinuous mentions in few-shot support set to train the model to handle these abnormal data.
>
> To extend our paper and demonstrate that our methods do have a potential to handle different NER settings in a unified framework, we added these two experiments under data abundant setting:
>
>  - We use T5-large as backbone and report span-level F1.
>  - Data abundant Nested NER:
>    - We conduct experiments on GENIA [2](Kim et al.,2003). We follow BARTNER [3](Yan et al.,2021) to use five entities types and split the train, dev, test as 8.1:0.9:1.0. The results are in the following Table.
>    - The result shows that in data abundant nested NER setting, our proposed method greatly outperforms BARTNER and other SOTA methods.
>    -
> | GENIA                                 | P     | R     | F     |
> |---------------------------------------|-------|-------|-------|
> | Li et al. (2020b)[BERT-Large] [4]     | 81.25 | 76.36 | 78.72 |
> | Yu et al. (2020)[BERT-Large] [5]      | 79.43 | 78.32 | 78.87 |
> | Wang et al. (2020a)[BERT-Large] [6]   | 79.45 | 78.94 | 79.19 |
> | BARTNER [3]                           | 78.87 | 79.6  | 79.23 |
> | 2INER (Main-Task + ET + EE + TE + EX) | 82.9  | 80.74 | 81.81 |
>
>  - Data abundant Discontinuous NER:
>    - We conduct experiments on CADEC [1](Karimi et al.,2015). Following BARTNER [3](Yan et al.,2021), since only the Adverse Drug Events (ADEs) entities include discontinuous data, only these entities were considered.
>    - The result shows that in data abundant discontinuous NER setting, our proposed method greatly outperforms BARTNER and other SOTA methods.
>    -
> | CADEC                                 | P     | R     | F     |
> |---------------------------------------|-------|-------|-------|
> | Metke-Jimenez and Karimi (2016) [7]   | 64.4  | 56.5  | 60.2  |
> | Tang et al. (2018) [8]                | 67.8  | 64.9  | 66.3  |
> | Dai et al. (2020)[ELMo] [9]           | 68.9  | 69.0  | 69.0  |
> | BARTNER [3]                           | 70.08 | 71.21 | 70.64 |
> | 2INER (Main-Task + ET + EE + TE + EX) | 71.18 | 75.26 | 73.16 |
>
> Because we do have difficulty in finding comparison for few-shot nested / discontinuous setting and other reasons above, we didn't add few-shot experiments.
>
> Thank you for mentioning it, and we will add these experiments as well as our revision for our introduction to our paper.
>
>
> > Type Extracting -> Type Extraction
>
> We admit that this is a grammar issues, and we have already modified this problem in our paper. Thank you for correcting us.
>
> > A Figure showing text-to-text NER in the introduction would be nice
>
> Thank you for your advice. And we have already made a new figure showing the text-to-text NER in our introduction. We also believe that adding a new figure can greatly make our paper easier to follow. Thanks a lot.
>
>
> References:
>
> [1]: Sarvnaz Karimi, Alejandro Metke-Jimenez, Madonna Kemp, and Chen Wang. 2015. Cadec: A corpus of adverse drug event annotations. J. Biomed. Informatics, 55:73–81.
>
> [2]: Jin-Dong Kim, Tomoko Ohta, Yuka Tateisi, and Jun’ichi Tsujii. 2003. GENIA corpus - a semantically annotated corpus for bio-textmining. In Proceedings of the Eleventh International Conference on Intelligent Systems for Molecular Biology, June 29 - July 3, 2003, Brisbane, Australia, pages 180–182.
>
> [3]: Hang Yan, Tao Gui, Junqi Dai, Qipeng Guo, Zheng Zhang, and Xipeng Qiu. 2021. A unified generative framework for various NER subtasks. In Proceedings of the 59th Annual Meeting of the Association for Computational Linguistics and the 11th International Joint Conference on Natural Language Processing(Volume 1: Long Papers), pages 5808–5822, Online. Association for Computational Linguistics.
>
> [4]: Xiaoya Li, Jingrong Feng, Yuxian Meng, Qinghong Han, Fei Wu, and Jiwei Li. 2020b. A unified MRC framework for named entity recognition. In Proceedings of the 58th Annual Meeting of the Association for Computational Linguistics, ACL 2020, Online, July 5-10, 2020, pages 5849–5859. Association for Computational Linguistics.
>
> [5]: Juntao Yu, Bernd Bohnet, and Massimo Poesio. 2020. Named entity recognition as dependency parsing. In Proceedings of the 58th Annual Meeting of the Association for Computational Linguistics, ACL 2020, Online, July 5-10, 2020, pages 6470–6476. Association for Computational Linguistics.
>
> [6]: Jue Wang, Lidan Shou, Ke Chen, and Gang Chen. 2020a. Pyramid: A layered model for nested named entity recognition. In Proceedings of the 58th Annual Meeting of the Association for Computational Linguistics, ACL 2020, Online, July 5-10, 2020, pages 5918–5928. Association for Computational Linguistics.
>
> [7]: Alejandro Metke-Jimenez and Sarvnaz Karimi. 2016. Concept identification and normalisation for adverse drug event discovery in medical forums. In Proceedings of the First International Workshop on Biomedical Data Integration and Discovery (BMDID 2016)co-located with The 15th International Semantic Web Conference (ISWC 2016), Kobe, Japan, October 17, 2016, volume 1709 of CEUR Workshop Proceedings. CEUR-WS.org.
>
> [8]: Buzhou Tang, Jianglu Hu, Xiaolong Wang, and Qingcai Chen. 2018. Recognizing continuous and discontinuous adverse drug reaction mentions from social media using LSTM-CRF. Wirel. Commun. Mob. Comput., 2018.
>
> [9]: Xiang Dai, Sarvnaz Karimi, Ben Hachey, and C´ecile Paris. 2020. An effective transition-based model for discontinuous NER. In Proceedings of the 58th Annual Meeting of the Association for Computational Linguistics, ACL 2020, Online, July 5-10, 2020, pages 5860–5870. Association for Computational Linguistics.

---

### Official Review · Reviewer_WRbf · 2023-08-11

**Soundness:** 4

**Excitement:**

3: Ambivalent: It has merits (e.g., it reports state-of-the-art results, the idea is nice), but there are key weaknesses (e.g., it describes incremental work), and it can significantly benefit from another round of revision. However, I won't object to accepting it if my co-reviewers champion it.

**Paper Topic And Main Contributions:**

In this work, the author proposed a new method for zero-shot named entity recognition using text sequence-to-sequence modeling and generates slot information in the input sentence as a sequence. In order to strengthen the model's ability to understand the semantic relation between entity labels and the sentence, the author proposed to train the model using a new auxiliary task called type extracting, which identifies what type of entities exist in the input sentence. Additionally, the proposed method concatenates demonstration example into the input sentence to incorporate external knowledge into the model. The authors showed that the proposed method significantly outperforms SOTA baselines in few-shot NER.

**Questions For The Authors:**

One can argue that type extraction can be inferred from entity typing. For example, given the sentence "are there any 4 star restaurant?". Entity typing gives the result "4 star is a rating", which immediately leads to "rating type exists in the sentence". As a result, it's not clear whether training with type extraction can can actually help the model to understand semantic information of the input sentence and the labels better.

**Reasons To Accept:**

The paper proposed a novel auxiliary task, called type extracting, to improve the NER model's ability to understand the semantic relations between the sentence and the labels.
The proposed method is the first that utilizes demonstration example and show that this method improve NER few-shot performance.

**Reasons To Reject:**

It is not clear how much type extraction contributes to the overall performance of the NER model. Even though the author provided ablation study in section 4.5 and show the results in Table 2, the results don't illustrate the improvement brought up by type extraction as an auxiliary task. It would be nice if the authors can show the result of 2INER vs 2INER+TE.
Demonstration example was shown to be effective for NER task in previous work, even though it is not sequence-to-sequence. Therefore, the result about improvement by demonstration example is not surprised.

**Reproducibility:**

5: Could easily reproduce the results.

**Reviewer Confidence:**

3: Pretty sure, but there's a chance I missed something. Although I have a good feel for this area in general, I did not carefully check the paper's details, e.g., the math, experimental design, or novelty.

---

> ### Author Rebuttal · Authors · 2023-08-28
>
> > It is not clear how much type extraction contributes to the overall performance of the NER model. Even though the author provided ablation study in section 4.5 and show the results in Table 2, the results don't illustrate the improvement brought up by type extraction as an auxiliary task. It would be nice if the authors can show the result of 2INER vs 2INER+TE.
>
> I think this problem is caused by a misunderstanding of the symbol in my article. And me myself also think that it may be confusing.
>
> In line-470, I explained that "The names InstructionNER and 2INER in the tables mean training with main-task data only". The intention of my statement is that the meanings of "InstructionNER" and "2INER" in the table are equivalent ("InstructionNER" == "2INER").
>
> In other words, the meaning of the symbol can be simplified as follows:
>
> | Symbol                       | Training Tasks Details                                                                |
> |------------------------------|---------------------------------------------------------------------------------------|
> | InstructionNER & 2INER       | Main-Task                                                                             |
> | 2INER+EX          | Main-Task + Demonstrate Example                                                       |
> | 2INER+TE          | Main-Task + Type Extraction                                                           |
> | 2INER+ET,EE,TE,EX | Main-Task + Entity Typing + Entity Extraction + Type Extraction + Demonstrate Example |
>
> Therefore, the suggestion of 2INER vs 2INER+TE (Main-Task vs Main-Task + Type Extraction) has already been done in Table 2 (which is InstructionNER vs 2INER+TE). So we can find out in Figure 2 that adding TE tasks on the Main-Task greatly increase F1 (red vs green bars). And because the Main-Task in our work is the same as those in InstructionNER paper, so we copy their results in Table 2.
>
> I'm very sorry that this error has caused you trouble. And we admit that "InstructionNER" == "2INER" is confusing, so this confusing symbols will be modified. Thank you for pointing that out!
>
> > Demonstration example was shown to be effective for NER task in previous work, even though it is not sequence-to-sequence. Therefore, the result about improvement by demonstration example is not surprised.
>
> Yes, you are right. Many previous works have shown the effectiveness of demonstration examples in other scenarios. And our work validates that it is equally effective in the text-to-text few-shot NER task.
>
> > One can argue that type extraction can be inferred from entity typing. For example, given the sentence "are there any 4 star restaurant?". Entity typing gives the result "4 star is a rating", which immediately leads to "rating type exists in the sentence". As a result, it's not clear whether training with type extraction can actually help the model to understand semantic information of the input sentence and the labels better.
>
> Thanks for your question, and we think that it's a worthy topic of discussion. This question is related to the intention of adding any auxiliary task.
>
> Firstly, every ability related to solving a specific downstream task can be learned through Main-Task. For example, when we train the model to predict "XXX span is a YYY entity", the model learns everything in order to solve this NER problems (which may include Entity Typing, Entity Extraction and Type Extraction, but we are not sure about that because those skills are achieved through hidden model parameters). Therefore, we believe that it is unreasonable to claim that auxiliary tasks are useless simply because they are implicit in the Main-Task. If it were true, then all auxiliary tasks (ET, EE, TE) would be useless. This is because the Main-Task includes all the necessary skills required to complete the NER.
>
> Secondly, the intention of adding auxiliary tasks is to artificially design thinking processes for completing the NER task and decompose it into several subtasks. This process is similar to human supervision, because it directly informs the model which sub-skills it needs to learn in order to complete the NER task. Furthermore, they visualize the model's hidden parameters into natural language to indicate that model itself do learn the process to complete NER, which increases the model's interpretability.
>
> Finally, we believe the only way to ensure the auxiliary task actually help the Main-Task is through observing the F1 score. Because auxiliary tasks (including Type Extraction) themselves are linguistically interpretable, as long as the final F1 score of NER is increasing, we believe that it is reasonable to claim the effectiveness of Type Extraction.

---

### Official Review · Reviewer_Cuzv · 2023-08-14

**Soundness:** 3

**Excitement:**

3: Ambivalent: It has merits (e.g., it reports state-of-the-art results, the idea is nice), but there are key weaknesses (e.g., it describes incremental work), and it can significantly benefit from another round of revision. However, I won't object to accepting it if my co-reviewers champion it.

**Paper Topic And Main Contributions:**

This paper proposes a novel text-to-text framework for Few-Shot Named Entity Recognition tasks, and introduces an auxiliary task called typing extracting. Also, this work tries to improve performance via in-context examples. The efficiency is validated on several open-sourced datasets.

**Reasons To Accept:**

1. a well structured work and easy to read
2. validated on multiple datasets and report detailed empirical results
3. the main method it-self seems to be good

**Reasons To Reject:**

1. the proposed auxiliary task kinda lack intrinsic novelty

**Reproducibility:**

4: Could mostly reproduce the results, but there may be some variation because of sample variance or minor variations in their interpretation of the protocol or method.

**Reviewer Confidence:**

3: Pretty sure, but there's a chance I missed something. Although I have a good feel for this area in general, I did not carefully check the paper's details, e.g., the math, experimental design, or novelty.

---

> ### Author Rebuttal · Authors · 2023-08-28
>
> > the proposed auxiliary task kinda lack intrinsic novelty
>
> We have to admit that the proposed auxiliary task only constructs the model's input-output at the linguistic level and does not involve mathematical reasoning. But we believe that with the popularity of text-to-text large language models, selecting the task that the model performs at the linguistic level, without changing the model structure, is the most lightweight and straightforward fine-tuning approach. Therefor, we believe that discussing what kind of auxiliary tasks can help downstream NER learning is valuable.
>
> And in terms of the novelty, we have abstracted the learning path of NER into three steps (instead of two), and demonstrated through experiments that our proposed auxiliary task can serve as a more fundamental NER ability, empowering both span recognition and entity labeling abilities.
>
> Furthermore, the proposed auxiliary task is efficient and easy to implement. By simply adding this auxiliary task, we were able to achieve significant improvement on downstream tasks, which also demonstrates the value of making linguistic modifications to model's input-output, despite their simplicity.

---

### Meta-Review · Area_Chair_de2W · 2023-09-19

**Recommendation:** 3

**Metareview:**

This paper proposes a text-to-text framework for few-shot named entity recognition by adopting some cases for in-context learning. The proposed method is not novel enough but shows the effectiveness on some open-domain scenarios. The experimental results have demonstrated the effectiveness of the proposed method. Although some reviewers have raised some concerns about some statements in the experiment, all of them generally agree with the point of the proposed method.

---

### Decision · Program_Chairs · 2023-10-07

**Decision:**

Accept-Findings

**Comment:**

This paper proposes a text-to-text framework for few-shot named entity recognition by adopting some cases for in-context learning. The proposed method is not novel enough but shows the effectiveness on some open-domain scenarios. The experimental results have demonstrated the effectiveness of the proposed method. Although some reviewers have raised some concerns about some statements in the experiment, all of them generally agree with the point of the proposed method.